# Rethinking Public Private Mix (PPM) Performance in the Tuberculosis Program: How Is Care Seeking Impacting This Model in High TB Burden Countries?

**DOI:** 10.3390/healthcare10071285

**Published:** 2022-07-12

**Authors:** Victor Abiola Adepoju, Olanrewaju Oladimeji, C. Robert Horsburgh

**Affiliations:** 1Department of HIV and Infectious Diseases, Jhpiego (An Affiliate of John Hopkins University), Abuja 900271, Nigeria; schrodinga05@yahoo.com; 2Department of Public Health, Faculty of Health Sciences, Walter Sisulu University, Mthatha 5117, South Africa; 3Section of Infectious Diseases, Boston Medical Center, Boston, MA 02118, USA; rhorsbu@bu.edu; 4Department of Biostatistics, Boston University School of Public Health, Boston, MA 02118, USA; 5Department of Global Health, Boston University School of Public Health, Boston, MA 02118, USA

**Keywords:** public–private mix, tuberculosis, risk profile, notification

## Abstract

In many high TB burden countries with enormous private-sector presence, up to 60–80% of the initial health-seeking behavior occurs in the private sector when people fall sick. Private-sector providers are also perceived to offer poorer-quality health service, and contribute to TB notification gaps and the spread of multidrug-resistant tuberculosis (MDR-TB). Recent efforts have focused on the expansion of TB services among private providers through public–private mix (PPM) initiatives. However, whether such efforts have matched the contribution of the private sector in TB notification, considering its enormous health-seeking volume, is debatable. Here, we argue that evaluating PPM program performance on the basis of the proportion of private-sector health seeking and level of undernotification is an imperfect approach due to differentials in tuberculosis risk profiles and access among patient populations seeking private care when compared with the public sector. We suggest a uniform definition of what constitutes PPM, and the standardization of PPM reporting tools across countries, including the ability to track patients who might initially seek care in the private sector but are ultimately publicly notified. PPM programs continue to gain prominence with rapid urbanization in major global cities. A universal health coverage framework as part of the PPM expansion mandate would go a long way to reduce the catastrophic cost of seeking TB care.

## 1. Background

In many places of the world, a massive proportion of tuberculosis (TB) patients seek care first from the private sector when not feeling well. In Nigeria, up to 60% of patients’ first interactions occur in a private healthcare setting, while this could be as high as 80% in Southeast Asia [1,2]. In Myanmar, 73% of TB patients initially visited the private sector, although many of them ended up being publicly notified [3]. Studies on the quality of TB services in the public and private sectors, and the impact on provider choice have mixed findings depending on the dimensions of quality being assessed [4]. Patients often choose private providers because of the proximity and convenience of locations, greater privacy, the flexibility of operation hours, and the perception of quality, for instance, reduced waiting times and better patient–provider interactions in the private sector [4]. This presents opportunities and challenges to tuberculosis control efforts in the private sector. It has been suggested that the private sector, particularly when not engaged by the National Tuberculosis Program (NTP), provides inappropriate diagnostic tests, substandard, non-quality-assured medications that are often expensive and paid out of pocket. This results in poor treatment outcomes and, in some cases, the development of multidrug-resistant tuberculosis (MDR-TB). In addition, the private sector often fails to report and notify the National Tuberculosis Program (NTP) surveillance systems. Public–private mix (PPM) collaboration is feasible and cost-effective if well-implemented on a large scale and patient-centered [5]. The need to engage the private sector, and strengthen the quality of care and TB notification systems is more crucial now than ever.

## 2. PPM Expansion Unmatched with Private-Sector Incident TB Notification

The private sector has lately received attention with the expansion of TB services among its providers. India and Nepal are among the few countries with documented evidence of increased case notification from PPM, but success remains distant and unmatched with the scale of engagement. For instance, six PPM projects in India that followed up for a median duration of 18 months reported a 15% (2–26%) contribution to overall case notification [6]. In Myanmar, PPM contributed 15–17% to childhood TB notification [7], 11% contribution was reported from Nigeria [8], and in Pakistan, PPM contributed 17% of the bacteriological diagnosed pulmonary TB cases [9]. The training of private health providers in Vietnam also led to a 7% increase in overall case notifications [10]. Among the Big Seven countries that contributed up to 60% of global TB notification gaps, the annual number of notifications associated with PPM in these seven countries increased from 225,000 cases in 2010 to more than 1.8 million cases in 2019 [11]. The proportion of total notifications contributed by the public–private mix in these countries also increased from 10% to nearly 30% in the same period [11]. In the leading country, Bangladesh, PPM contributed 30% to TB case notification, followed by India and Pakistan, where PPM contributed 21% in both countries. However, relative to estimated incidence, notification was the highest in Bangladesh (18%), followed by Pakistan (14%), India (14%) and Myanmar (10%). These figures were far from comparable with private-sector health-seeking behavior in these countries [11].

## 3. Measurement of PPM Program Performance Is Unclear

The challenges of measuring the contribution and performance of PPM programs are the disparity in what constitutes PPM, the limitations of how PPM targets are set, and variability in the ways in which performance is measured in countries and intervention projects. The World Health Organization (WHO) defines PPM as the engagement of private-sector providers of TB care that include individual and institutional private providers, the corporate or business sector, nongovernmental organizations (NGOs), faith-based organizations (FBOs), and mission hospitals [11]. PPM also includes numerous informal providers, drug shops, and qualified independent providers [8]. How countries report these providers has an enormous impact on private-sector notification and progress assessment. For instance, until recently, some countries reported the activities of patent proprietary medicine vendors (PPMVs), community pharmacists (CPs), and stand-alone laboratories as “community referral”, which often ends up in public notification figures [8]. The standardization of PPM definition and PPM reporting tools across countries is needed for the objective assessment of private-sector impact on the overall TB case notification.

Two approaches have been used to set targets and measure the performance of PPM programs, namely, the inventory study approach and the approach that multiplies the estimated TB incidence by the share of private-sector primary-health-seeking behavior [8]. Inventory study estimates the contribution of the private sector with the formula c (1−U) = +U, where c is the current PPM share of total notifications, and U is the level of under-reporting [8]. There are fundamental problems with measuring PPM performance or setting targets for PPM contribution using these approaches. The inventory study approach fails to consider the considerable proportion of patients who, though they had initially visited the private sector, ended up being publicly notified TB patients. For instance, in Myanmar, up to 73% of TB patients initially visited private general practitioner (GP) clinics before presenting to public TB centers due to the perception that the private sector is fee-paying, and free TB treatment could only be accessed in the public sector [3]. Estimating PPM contribution or setting PPM target by multiplying TB incidence with the private-sector share of primary-health-seeking behavior also presents flaws, as private-sector patients may differ in TB risk factor profile and symptomatology compared with those seeking care in the public sector. This has an enormous impact on the chances of TB detection and notification in both sectors when all other factors are held constant.

## 4. Tuberculosis Risk Profiles among Health Seekers as the Yardstick for PPM Program Evaluation

Inherent risk predicts TB patient choice, and the selection of private and public facilities [12,13]. In South Korea, age, sex marital status, and education predicted the selection of public hospital [13]. In Ghana, age > 80 years, higher education, and wealth predicted the use of private hospitals [14]. One report also noted that private hospitals have a lower infection rate and largely manage patients with a lower risk of infection when compared with public hospitals [15].

Setting-specific risks for TB can be evaluated using the number needed to screen (NNS), which is the number required to screen to identify one TB case in a population. This could ultimately give an idea of the differences in TB identified from both sectors. NNS measures the efficacy of TB screening intervention and potential to increase TB case notification in a given population. Although NNS studies comparing public and private providers are lacking, private GPs in Pakistan needed to screen 524 patients to identify one bacteriologically positive TB patient, while government health facilities in India needed to screen 328 patients to identify one bacteriologically positive TB patient [16,17,18]. In addition, only 3.3% of outpatient department (OPD) attendees in Indian private hospitals and clinics had clinical TB, compared to 4.1–4.5% of the attendees in the public hospital [19]. In the low TB incident city of Victoria, Australia, a study compared tuberculosis management under public and private healthcare providers in 2002–2015, and found that patients attending private settings presented an earlier symptom onset, and had fewer positive sputa and less frequent abnormal radiology for extrapulmonary TB, although they had a poorer patient assessment and longer delay for TB treatment when compared with public-sector patients [20]. This indicates differing disease severity among patient cohorts presenting in public and private healthcare settings. The above findings also explain the ‘notification dormancy’ of certain private hospitals and clinics years after engagement by NTP despite efforts to clinically screen OPD attendees in many instances. For instance, in Nigeria, a review noted that 30% of engaged private for-profit (PFP) clinics and 12% of faith-based organizations (FBOs) registered no TB cases. In comparison, 6% and 16% of FBOs and PFPs, respectively, registered only 1 TB case in 2017 [8]. The authors added that 80% of all notifications came from 61 facilities (26% of the total engaged). While the concentration of TB cases among a relatively small percentage of providers is not unexpected, the vast number of dormant providers suggests the necessity for strategic PPM re-engagement and prioritization based on the levels of care, provider type, and geographic locations of the high-risk population.

Previously reported risk factors for tuberculosis include malnutrition, low family income and poverty, alcohol, smoking, the male gender, poor housing, low education and illiteracy, diabetes, HIV, family history of TB, overcrowding, a lack of insurance, and the absence of BCG [21,22,23,24]. In India, the prevalence of diabetes, a major risk factor for TB, was higher among patients attending public hospitals and clinics (9.24%) compared with private hospital patients (8.02%) [19]. In Pakistan, 70.6% of public tertiary hospital patients smoked, compared with 29.4% in private hospitals [25]. Income also influences the choice of providers in India and South Korea. In India, those earning less than INR 2000 monthly sought care in the government-owned setting, while those earning more than INR 2000 per month sought care in the private sector. In addition, 97% of OPD attendees in the highest income bracket of income chose a private hospital in South Korea [13,26]. Having to pay for expensive medical injections in South Korea was associated with the use of public hospitals, which further supported the notion that the choice of providers for chronic diseases such as tuberculosis is influenced by the economic situation of individuals [13].

Moreover, in South Korea and Ghana, a higher proportion of individuals with health insurance patronized public hospitals, citing that some private hospitals could not be assessed with the government-funded National Health Insurance Scheme (NHIS) [12,13]. Australian patients had lower health literacy questionnaire (HLQ) scores. However, this was not associated with the use of public hospital services [27] while education correlated with the use of chemist shops in Ghana [14]. In Aurangabad, India, a higher proportion of those with a lower qualification than a high-school diploma visited a private provider. In contrast, those with higher qualifications visited public providers [28], and in South Korea, individuals with a middle school education or less were more likely to choose public rural provinces than those living in cities or with a higher education, who were less likely to seek treatment at a public hospital. [13]. Malnutrition is a significant risk factor for TB. In Kenya, a higher proportion of public patients had a body mass index (BMI) >18.5 mg/m^2^, while the proportion of patients treated for malnutrition in the private sector was lower [29]. The authors further noted that the number of previously treated TB patients was significantly higher in the public sector [29]. In Ghana, an age of more than 80 years also predicted the use of private hospitals [14]. The global tuberculosis report has consistently reported higher numers of TB cases among men than those in women [30]. Reports from Aurangabad, India showed that more females than males self-medicated and sought care in the private sector; in South Korea, more men (6.36%) than women (5.11%) patronized public hospitals [13,28]. This highlights opportunities for finding more TB cases in the public sector, which a considerable proportion of men patronized.

Although 73% of TB patients in Myanmar initially sought care in the private sector, many considered the private facilities to be fee-paying. These patients accessed TB treatment in the public sector by switching from private to public facilities due to the financial constraints of having to pay for drugs and consultations, and on the basis of advice given by ex-TB patients and relatives [3]. These underscore the importance of eliminating the catastrophic costs of TB to improve real access to TB services and notification in the private sector. Many patients who initially seek care in the private sector often switch to the public sector for TB treatment, but remain in care for other services within their family-centered private GPs despite receiving TB treatment in the public sector. This further points to the opportunities and challenges associated with PPM programs. The differentials in tuberculosis risk profiles and access among private- and public-sector attendees, and the dynamics of private-to-public sector mobility of patients who initially sought care in the private sector need to be considered in measuring performance and setting targets for PPM programs.

In contrast, studies from Vietnam found no significant difference in the socioeconomic status and use of private physicians of TB or suspected TB patients, at least prior to TB diagnosis [31]. That notwithstanding, the above findings underscore the different socioeconomic dynamics of patients patronizing the public and private sectors that limit their access to TB services. Although the private sector offers services with shorter waiting times and better provider–patient interaction, patients seeking care in this sector appear to carry milder tuberculosis risk profiles and limited access. For instance, they are less likely to be malnourished, uninsured, uneducated, diabetic, and from a low-income class. Depending on the context, settings, and status of engagement of the private provider with NTP, even when those with higher risk profiles attended the private sector, access to tuberculosis services was limited by the financial obligations of having to pay for consultation, diagnosis, and drugs, which redirected their care-seeking pathways to the public sector, where they were ultimately treated and notified. In addition to TB risk profiles and access, the perception of providers has a strong influence on choice and varies across settings. For instance, in China, patients believed that even less-qualified providers could handle TB patients [32], while in Bangladesh, patients believed that they could only receive quality TB treatment from primary health centers that were not necessarily linked with NTP [33]. Surprisingly, in India, patients patronizing private hospitals believed that free TB medications in the public sector were ineffective [34]. These are settings where specific considerations on how the perceptions of suspected or actual TB patients about different categories of private and public providers are critical for the rapidly expanding PPM programs. One of the limitations of this study is the scarcity of comparative data on TB risk profiles among public- and private-sector patients in high-incident countries. This weakness limits the ability to produce a definite conclusion regarding the study findings.

## 5. Insights and Implications

We found heterogeneity in how tuberculosis public–private mix (PPM) was defined, and a lack of harmonized data collection tools and performance standards, which renders cross-country performance assessment challenging.The private-to-public sector mobility of TB patients and sectoral differences in socioeconomic risk profile highlights the inadequacies of using tuberculosis health-seeking behaviors to set the targets and measure performance of PPM programs.Global consensus on what constitutes PPM TB care providers, harmonization of reporting tools, and re-evaluation of how PPM program performance is evaluated and compared across countries.Policies that integrate universal health insurance into tuberculosis PPM expansion framework to limit catastrophic TB costs and other socioeconomic indices that are most likely higher among private-sector health seekers.

## 6. Conclusions and Policy Recommendations

The results highlight the difficulties in measuring PPM program performance across countries due to variations in what constitutes PPM, and inadequacies in using health-seeking behavior as a performance benchmark. The findings also suggest that, although a huge proportion of initial health seeking takes place in the private sector, symptomatic patients visit private-sector providers earlier, carry a lesser TB risk profile, and are less likely to have positive sputum specimens. In other instances, certain subgroups of initial private-sector health seekers ended up in the public sector for TB treatment as they were not adequately assessed or promptly commenced on treatment by private providers.

Tuberculosis risk often has its roots in poverty among high-risk populations who in many cases have opted for the cheaper public-sector alternative; hence, we propose an integrated universal health insurance plan to be integrated into tuberculosis PPM expansion frameworks to limit the catastrophic cost of TB. We recommend a scoping review or comparative analysis that would examine the relationship between socioeconomic status (educational level, social class, income, etc.) and health-service utilization patterns for TB services in public and private hospital settings. Health inequity may possibly favor or adversely select a particular socioeconomic group. The suggested studies help in unraveling patient selection types in both the private and public sectors, and these could further help in adjusting PPM performance expectations. Of importance is the use of a common definition of what constitutes PPM, and the harmonization of country data collection tools to include reporting standards of patients who had initially sought care in the private sector, but ended up being notified in the public sector. Each country needs to evaluate the various private-sector actors and determine how best to regulate and scale up expansion among them. Lastly, the present study drew attention to how patient perception regarding various qualified and unqualified private TB service providers varied across contexts and settings. Context-specific health promotion campaigns are needed to target high TB risk, low-income communities, dispel myths about TB, and emphasize the availability and quality of TB services in both formal and informal private provider settings.

## Data Availability

Not applicable.

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
