# Peer review of "Rethinking Public Private Mix (PPM) Performance in the Tuberculosis Program: How Is Care Seeking Impacting This Model in High TB Burden Countries?"

_healthcare, 2022, doi:10.3390/healthcare10071285_

Round 1

Reviewer 1 Report

Overall, the authors have provided substantial arguments to challenge the standards of measurement for the performance of the PPM program. Some minor corrections required to be addressed are listed below. 

Page 1 : Line 42 - NTP hasn't been explained before usage

Page 2 : Line 61-62 the data presented by the authors for comparison is not as per the latest WHO Global TB report. Authors should consider updating the information as per the latest statistics as well as update reference  11. 

Line 92 and 113: Authors have used the abbreviation before its definition which is on line 177. 

Line 154, it would be nice to mention the country in which Aurangabad is located while comparing it with other countries.

Referencing style is not in one format and some of the references cited in the paper are not reflecting current statistics. 

Author Response

Dear Editor,

We thank the Handling Editor and reviewer for their efforts. We are pleased that there is interest in this topic and are grateful for the constructive comments that have improved the manuscript.

We have addressed each of the reviewers’ comments in a point-by-point manner in the table below. The reviewers’ comments are on the left column, and our responses are highlighted in blue font on the right. We have also made corresponding changes in the manuscript, indicating the revised sections with "track changes. While hoping that these changes will meet with your favorable consideration, we hold ourselves at your disposition for any further clarifications.

S/N

Comments

Response

1

Page 1 : Line 42 - NTP hasn't been explained before usage

NTP now spelt out as National Tuberculosis Program. Page 1, Line 42

2

Page 2 : Line 61-62 the data presented by the authors for comparison is not as per the latest WHO Global TB report. Authors should consider updating the information as per the latest statistics as well as update reference  11. 

This is now replaced with citation from the 2020 Global TB Report and the citation 11 has also been replaced as well in the reference list. Page 2, Line 60-64

3

Line 92 and 113: Authors have used the abbreviation before its definition which is on line 177. 

Full meaning of GP  now defined for the first time in line 97, while GP maintained as an abbreviation in Line 182

4

Line 154, it would be nice to mention the country in which Aurangabad is located while comparing it with other countries.

India, the country where Aurangabad was located is now indicated on line 159

5

Referencing style is not in one format and some of the references cited in the paper are not reflecting current statistics. 

Thank you very much. We have changed some quoted statistics to current figure e.g Page 2, line 60-64. All references have now been revised to align with MDPI reference style

Regards,

--------------------------

Olanrewaju Oladimeji II MB; BS, MSc, MPA, FRSPH, PhD
Research Chair: Department of Public Health, Walter Sisulu University, Eastern Cape, South Africa

Visiting Scholar: Harvard T.H. Chan School of Public Health, Boston, USA

Adjunct Professor: Faculty of Health Sciences, Durban University of Technology, South Africa.

email: [email protected]@wsu.ac.za

Reviewer 2 Report

It is very interesting to present this option in the TB program. In my country, Colombia, there is no such link between the private and public sectors for the treatment of patients, since medicines can only be dispensed by the public programme.

For this reason I think it is convenient to explain much better how this mixed program works, the public sector supplies the medicines to the private so that the patient does not pay for them? Who covers the costs of laboratory tests?

How they can see this information is necessary for those people who do not know about this program.

Thanks 

Author Response

Point-by-Point response to the reviewer comments

Dear Editor,

We thank the Handling Editor and reviewer for their efforts. We are pleased that there is interest in this topic and are grateful for the constructive comments that have improved the manuscript.

We have addressed each of the reviewers’ comments in a point-by-point manner in the table below. The reviewers’ comments are on the left column, and our responses are highlighted in blue font on the right. We have also made corresponding changes in the manuscript, indicating the revised sections with "track changes. While hoping that these changes will meet with your favorable consideration, we hold ourselves at your disposition for any further clarifications.

S/N

Comments

Response

1

It is very interesting to present this option in the TB program. In my country, Colombia, there is no such link between the private and public sectors for the treatment of patients, since medicines can only be dispensed by the public programme.

Thanks very much for your positive feedbacks and for letting us know how the public and private sector TB service provision is organized in your country

2

For this reason, I think it is convenient to explain much better how this mixed program works, the public sector supplies the medicines to the private so that the patient does not pay for them?

Thanks for your suggestion. Whether clients pay for drugs diagnosis is a function of context and setting as well as engagement status of the private facility with NTP. Engaged private facilities in mots countries receive free anti-TB given free of charge to TB patients, ditto for genexpert while for unengaged, clients pay for loose, non-FDC drugs purchased by the private provider. In some countries like India, private sector clients pay for diagnosis irrespective of engagement status of the private provider. This has been added in line 195-196

3

Who covers the costs of laboratory tests?

Again this depend on the context/setting. In Nigeria where the international donors cover for diagnosis test and drugs, lab test like genexpert and drugs are free in the private sector while in India, some private providers purchased the Xpert machine and clients pay for the test. However, microscopy test is usually being paid for in private sector.

4

How they can see this information is necessary for those people who do not know about this program.

Yes, thanks for these feedbacks. We do hope that additions based on above comment will help the readers

Regards,

--------------------------

Olanrewaju Oladimeji II MB; BS, MSc, MPA, FRSPH, PhD
Research Chair: Department of Public Health, Walter Sisulu University, Eastern Cape, South Africa

Visiting Scholar: Harvard T.H. Chan School of Public Health, Boston, USA

Adjunct Professor: Faculty of Health Sciences, Durban University of Technology, South Africa.

email: [email protected]@wsu.ac.za

Reviewer 3 Report

The perspective "Rethinking Public Private Mix (PPM) performance in the Tu-2 berculosis program: How is the care-seeking impacting on this 3 model in high TB burden countries?" submitted by Victor et al is interesting to the people working in this area and is suitable for publication in Healthcare journal. However, the authors should make some modifications.

1. Abstract and conclusion needs to be concised

2. Minor grammatical errors must be corrected.

3. More statistical data has to be included for better understanding and getting a broad impression on the topic.

Author Response

Point-by-Point response to the reviewer comments

Dear Editor,

We thank the Handling Editor and reviewer for their efforts. We are pleased that there is interest in this topic and are grateful for the constructive comments that have improved the manuscript.

We have addressed each of the reviewers’ comments in a point-by-point manner in the table below. The reviewers’ comments are on the left column, and our responses are highlighted in blue font on the right. We have also made corresponding changes in the manuscript, indicating the revised sections with "track changes. While hoping that these changes will meet with your favorable consideration, we hold ourselves at your disposition for any further clarifications.

S/N

Comments

Response

1

The perspective "Rethinking Public Private Mix (PPM) performance in the Tu-2 berculosis program: How is the care-seeking impacting on this 3 model in high TB burden countries?" submitted by Victor et al is interesting to the people working in this area and is suitable for publication in Healthcare journal. However, the authors should make some modifications.

Thanks very much for your kind feedbacks and we are open to making changes suggested

2

Abstract and conclusion needs to be concised

Thanks for your feedbacks. We have looked at the abstract and conclusion section. The word count for the abstract is 215 which is still within word limit while the conclusion is a little longer as we combined the section with policy recommendations

3

Minor grammatical errors must be corrected.

Thanks. We have corrected these grammatical errors all through the document e.g Lines 125,201,207,231,215,216,226 etc

4

More statistical data has to be included for better understanding and getting a broad impression on the topic.

Thank you very much for the feedback. All relevant statistics have been included to support our arguments

Regards,

--------------------------

Olanrewaju Oladimeji II MB; BS, MSc, MPA, FRSPH, PhD
Research Chair: Department of Public Health, Walter Sisulu University, Eastern Cape, South Africa

Visiting Scholar: Harvard T.H. Chan School of Public Health, Boston, USA

Adjunct Professor: Faculty of Health Sciences, Durban University of Technology, South Africa.

email: [email protected]@wsu.ac.za
